# SOCIAL-TRANSMOTION: PROMPTABLE HUMAN TRAJECTORY PREDICTION

**Saeed Saadatnejad**[*], **Yang Gao**[*], **Kaouther Messaoud**, **Alexandre Alahi**
Visual Intelligence for Transportation (VITA) laboratory
EPFL, Switzerland
`{firstname.lastname}@epfl.ch`

## ABSTRACT

Accurate human trajectory prediction is crucial for applications such as autonomous vehicles, robotics, and surveillance systems. Yet, existing models often fail to fully leverage the non-verbal social cues human subconsciously communicate when navigating the space. To address this, we introduce *Social-Transmotion*, a generic Transformer-based model that exploits diverse and numerous visual cues to predict human behavior. We translate the idea of a prompt from Natural Language Processing (NLP) to the task of human trajectory prediction, where a prompt can be a sequence of x-y coordinates on the ground, bounding boxes in the image plane, or body pose keypoints in either 2D or 3D. This, in turn, augments trajectory data, leading to enhanced human trajectory prediction. Using masking technique, our model exhibits flexibility and adaptability by capturing spatiotemporal interactions between agents based on the available visual cues. We delve into the merits of using 2D versus 3D poses, and a limited set of poses. Additionally, we investigate the spatial and temporal attention map to identify which keypoints and time-steps in the sequence are vital for optimizing human trajectory prediction. Our approach is validated on multiple datasets, including JTA, JRDB, Pedestrians and Cyclists in Road Traffic, and ETH-UCY. The code is publicly available: https://github.com/vita-epfl/social-transmotion.

## 1 INTRODUCTION

Predicting future events is often considered an essential aspect of intelligence (Bubic et al., 2010). This capability becomes critical in autonomous vehicles (AV), where accurate predictions can help avoid accidents involving humans. Human trajectory prediction serves as a key component in AVs, aiming to forecast the future positions of humans based on a sequence of observed 3D positions from the past. It has been proven effective in the domains of autonomous driving (Saadatnejad et al., 2022a), socially-aware robotics (Chen et al., 2019), and planning (Flores et al., 2018; Luo et al., 2018). Despite acknowledging the inherent stochasticity that arises from human free will, most traditional predictors have limited performance, as they typically rely on a single cue per person (*i.e.*, their x-y coordinates on the ground) as input.

In this study, we explore the signals that humans consciously or subconsciously use to convey their mobility patterns. For example, individuals may turn their heads and shoulders before altering their walking direction—a visual cue that cannot be captured using a sequence of spatial locations over time. Similarly, social interactions may be anticipated through gestures like hand waves or changes in head direction. Our goal is to propose a generic architecture for human trajectory prediction that leverages additional information whenever they are available (*e.g.*, the body pose keypoints). To this end, in addition to the observed trajectories, we incorporate the sequence of other observed cues as input, to predict future trajectories, as depicted in Figure 1. We translate the idea of prompt from Natural Language Processing (NLP) to the task of human trajectory prediction, embracing all signals that humans may communicate in the form of various prompts. A prompt can be a sequence of x-y coordinates on the ground, bounding boxes in the image plane, or body pose keypoints in either 2D or 3D. We refer to our task as *promptable human trajectory prediction*. Nonetheless, effectively

---

[*] Equal contribution.

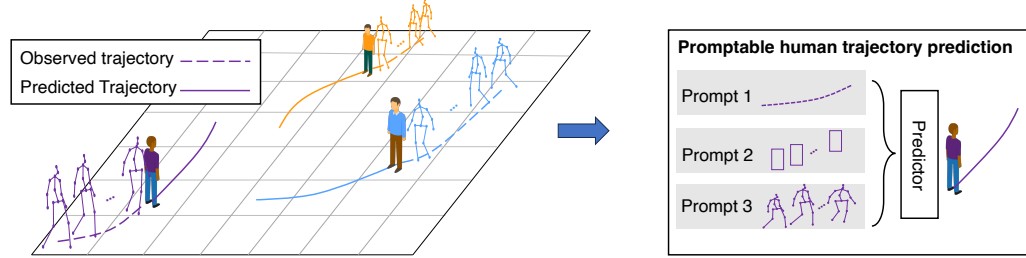

Figure 1: We present the task of *promptable human trajectory prediction*: Predict human trajectories given any available prompt such as past trajectories or body pose keypoints of all agents. Our model dynamically assesses the significance of distinct visual cues of both the primary and neighboring agents and predicts more accurate trajectories.

encoding and integrating all these diverse visual cues into the prediction model is significantly challenging.

We introduce *Social-Transmotion*, a generic and adaptable Transformer-based model for human trajectory prediction. This model seamlessly integrates various types and quantities of visual cues, thus enhancing adaptability to diverse data modalities and exploiting rich information for improved prediction performance. Its dual-Transformer architecture dynamically assesses the significance of distinct visual cues of both the primary and neighboring agents, effectively capturing relevant social interactions and body language cues. To ensure the generality of our network, we employ a training strategy that includes selective masking of different types and quantities of visual cues. Consequently, our model exhibits robustness even in the absence of certain visual cues. For instance, it can make predictions without relying on bounding boxes when pose information is unavailable, or it can use only trajectory inputs when no visual cues are accessible.

Our experimental results demonstrate that Social-Transmotion outperforms previous models on several datasets. Additionally, we provide a comprehensive analysis of the usefulness of different visual representations, including 2D and 3D body pose keypoints and bounding boxes, for the trajectory prediction task. We show that 3D pose keypoints more effectively capture social interactions, while 2D pose keypoints can be a good alternative when 3D pose information is unavailable. Furthermore, we explore the necessity of having poses of all humans in the scene at every time-step, as well as the necessity of 3D versus 2D poses or even just bounding boxes. In some applications, only the latter may be available. We provide an in-depth analysis of these factors in Section 4.

In summary, our contribution is twofold. First, we present Social-Transmotion, a generic Transformer-based model for promptable human trajectory prediction, designed to flexibly utilize various visual cues for improved accuracy, even in the absence of certain cues. Second, we provide an in-depth analysis of the usefulness of different visual representations for trajectory prediction.

## 2  RELATED WORKS

### 2.1  HUMAN TRAJECTORY PREDICTION

Human trajectory prediction has evolved significantly over the years. Early models, such as the Social Force model, focused on the attractive and repulsive forces among pedestrians (Helbing & Molnar, 1995). Later, Bayesian Inference was employed to model human-environment interactions for trajectory prediction (Best & Fitch, 2015). As the field progressed, data-driven methods gained prominence (Alahi et al., 2016; Gupta et al., 2018; Giuliari et al., 2021; Kothari et al., 2021; Monti et al., 2021; Sun et al., 2022; Zhang et al., 2019; Mangalam et al., 2021; Chen et al., 2023), with many studies constructing human-human interactions (Alahi et al., 2016; Kothari et al., 2021; Monti et al., 2021; Zhang et al., 2019) to improve predictions. For example, Alahi et al. (2016) used hidden states to model observed neighbor interactions, while Kothari et al. (2021) proposed the directional grid for better social interaction modeling. In recent years, researchers have expanded the scope of social interactions to encompass human-context interactions (Best & Fitch, 2015; Coscia et al., 2018; Sun et al., 2022) and human-vehicle interactions (Bhattacharyya et al.,

2021; Zhang & Berger, 2022). Various architectural models have been used, spanning from recurrent neural networks (RNNs) (Alahi et al., 2016; Salzmann et al., 2020), generative adversarial networks (GANs) (Gupta et al., 2018; Amirian et al., 2019; Hu et al., 2020; Huang et al., 2019) and diffusion models (Gu et al., 2022).

The introduction of Transformers and positional encoding (Vaswani et al., 2017) has led to their adoption in sequence modeling, owing to their capacity to capture long-range dependencies. This approach has been widely utilized recently in trajectory prediction (Yu et al., 2020; Giuliari et al., 2021; Li et al., 2022; Yuan et al., 2021) showing state-of-the-art performance on trajectory prediction (Girgis et al., 2022; Xu et al., 2023). Despite advancements in social-interaction modeling, previous works have predominantly relied on sequences of agents x-y coordinates as input features. With the advent of datasets providing more visual cues (Fabbri et al., 2018; Martin-Martin et al., 2021; Ionescu et al., 2014), more detailed information about agent motion is now available. Therefore, we design a generic Transformer-based model that can benefit from incorporating visual cues in a promptable manner.

## 2.2 Visual Cues for Trajectory Prediction

Multi-task learning has emerged as an effective approach for sharing representations and leveraging complementary information across related tasks. Pioneering studies have demonstrated the potential benefits of incorporating additional associated output tasks into human trajectory prediction, such as intention prediction (Bouhsain et al., 2020), 2D/3D bounding-box prediction (Saadatnejad et al., 2022b), action recognition (Liang et al., 2019) and respecting scene constraints (Sadeghian et al., 2019; Saadatnejad et al., 2021). Nevertheless, our focus lies on identifying the input modalities that improve trajectory prediction.

The human pose serves as a potent indicator of human intentions. Owing to the advancements in pose estimation (Cao et al., 2019), 2D poses can now be readily extracted from images. In recent years, a couple of studies have explored the use of 2D body pose keypoints as visual cues for trajectory prediction in image/pixel space (Yagi et al., 2018; Chen et al., 2020). However, our work concentrates on trajectory prediction in camera/world coordinates, which offers more extensive practical applications. Employing 2D body pose keypoints presents limitations, such as information loss in depth, making it difficult to capture the spatial distance between agents. In contrast, 3D pose circumvent this issue and have been widely referred to in pose estimation (Wandt et al., 2021), pose forecasting (Parsaeifard et al., 2021; Saadatnejad et al., 2023; 2024), and pose tracking (Reddy et al., 2021). Nevertheless, 3D pose data may not always be available in real-world scenarios. Inspired by a recent work in intention prediction that demonstrated enhanced performance by employing bounding boxes (Bouhsain et al., 2020), we have also included this visual cue in our exploration. Our goal is to investigate the effects of various visual cues, including but not limited to 3D human pose, on trajectory prediction.

Kress et al. (2022) highlighted the utility of an individual agent's 3D body pose keypoints for predicting its trajectory. However, our research incorporates social interactions with human pose keypoints, a feature overlooked in their study. Also, unlike Hasan et al. (2019), that proposed head orientation as a feature, we explore more granular representations. Our work enhances trajectory prediction precision by considering not only the effect of social interactions with human pose keypoints but also other visual cues. Moreover, our adaptable network can harness any available visual cues.

## 3 Method

We propose an adaptable model that effectively utilizes various visual cues alongside historical trajectory data in order to predict future human trajectories. We also recognize that different scenarios may present varying sets of visual cues. To address this, our model is designed to be flexible to handle different types and quantities of cues.

As illustrated in Figure 2, our model comprises two Transformers. Cross-Modality Transformer (CMT) takes as input each agent's previous locations and can incorporate additional cues such as the agent's 2D or 3D pose keypoints and bounding boxes from past time-steps. By incorporating these diverse cues, it generates a more informative representation of each agent's behavior. Then, Social Transformer (ST) is responsible for merging the outputs from the first Transformers of different

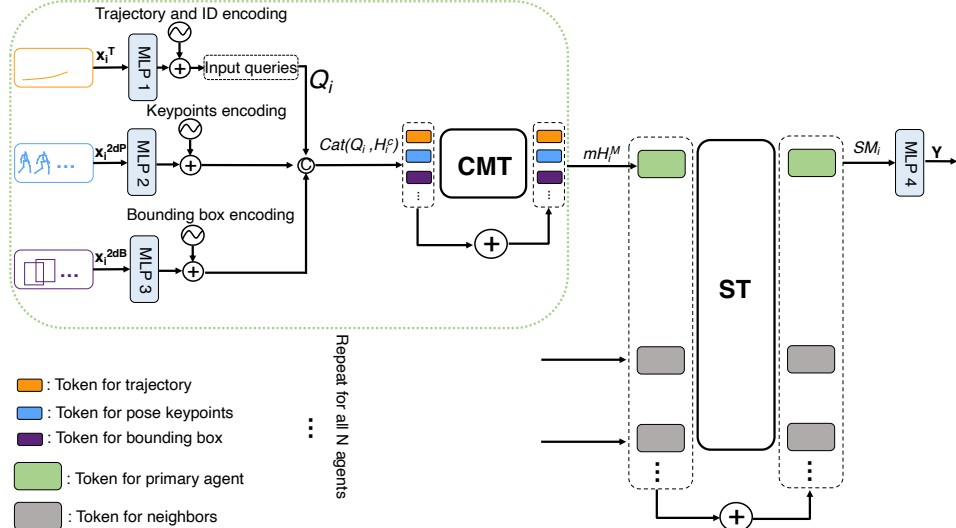

Figure 2: Social-Transmotion: A Transformer-based model integrating 3D human poses and other visual cues to enhance trajectory prediction accuracy and social awareness. Cross-Modality Transformer (CMT) attends to all cues for each agent, while Social Transformer (ST) attends to all agents' representations to predict trajectories.

agents. By combining these representations, ST captures interactions between agents, enabling the model to analyze their interplay and dependencies.

## 3.1 PROBLEM FORMULATION

We denote the trajectory sequence of agent $i$ as $x_i^T$, the 3D and 2D local pose sequences as $x_i^{3dP}$ and $x_i^{2dP}$ respectively, and the 3D and 2D bounding box sequences as $x_i^{3dB}$ and $x_i^{2dB}$, respectively. We also define the observed time-steps as $t = 1, ..., T_{obs}$ and the prediction time-steps as $t = T_{obs} + 1, ..., T_{pred}$. In a scene with $N$ agents, the network input is $X = [X_1, X_2, X_3, ..., X_N]$, where $X_i = \{x_i^c, c \in \{T, 3dP, 2dP, 3dB, 2dB\}\}$ depending on the availability of different cues. The tensor $x_i^c$ has a shape of $(T_{obs}, e^c, f^c)$, where $e^c$ represents the number of elements in a specific cue (for example the number of keypoints) and $f^c$ denotes the number of features for each element.

Without loss of generality, we consider $X_1$ as the primary agent. The network's output, $Y = Y_1$, contains the predicted future trajectory of the primary agent, following the standard notation.

## 3.2 INPUT CUES EMBEDDINGS

To effectively incorporate the visual cues into our model, we employ a cue-specific embedding layer to embed the coordinates of the trajectory and all visual cues for each past time-step. In addition, we utilize positional encoding techniques to represent the input cues' temporal order. We also need to encode the identity of the person associated with each cue and the keypoint type for keypoint-related cues (*e.g.*, neck, hip, shoulder). To tackle this, we introduce three distinct embeddings: one for temporal order, one for person identity, and one for keypoint type. The temporal order embedding facilitates the understanding of the sequence of cues, enabling the model to capture temporal dependencies and patterns. The person identity embedding allows the model to distinguish between different individuals within the input data. Lastly, the keypoint type embedding enhances the model's ability to extract relevant features and characteristics associated with different keypoint types movement. These embeddings are randomly initialized and learned during the training.

$$H_i^c = MLP^c(\mathbf{x_i^c}) + P,$$

where $MLP^c$ refers to cue-specific Multi-Layer Perceptron (MLP) embedding layers, and the tensor $P$ contains positional encoding information. The resulting tensor $H_i^c$ has a shape of $(T_{obs}, e^c, D)$, with $D$ representing the embedding dimension.

### 3.3 Latent Input Queries

We equip each agent with a set of latent queries, denoted as $Q_i$, which have the shape $(T_{pred} - T_{obs}, D)$. These latent queries are designed to encapsulate the motion information of each agent, given the substantial and variable quantity of input modalities. CMT encodes these queries. Subsequently, along with the representations of past trajectories, they are processed by ST.

### 3.4 Cross-Modality Transformer (CMT)

CMT in our model is designed to process various input embeddings. By incorporating different cues, CMT is capable of providing a comprehensive and informative representation of each agent's motion dynamics. Furthermore, CMT employs shared parameters to process the various modalities and ensure efficient information encoding across different inputs.

$$mQ_i, mH_i^c = \mathbf{CMT}(Q_i, H_i^c, c \in \{T, 3dP, 2dP, 3dB, 2dB\}).$$

In short, $\mathbf{CMT}$ transforms the latent representation of agent trajectory, $concat(Q_i, H_i^T)$, into a motion cross-modal tensor $mH_i^M = concat(mQ_i, mH_i^T)$ with shape $(T_{pred}, D)$, which is the concatenation of cross-modal trajectory tokens $mH_i^T$ and cross-modal latent queries $mQ_i$. All other cues' embedding tensors $H_i^c$ are mapped to $mH_i^c$ with shape $(T_{obs}, e^c, D)$.

It is important to note that while CMT receives inputs from various cues, only the motion cross-modal tensor $mH_i^M$ is passed to ST. Therefore, the number of input vectors to ST is independent of the number of the input cues. This decision is based on the assumption that the motion cross-modal features capture and encode information from various available cues.

### 3.5 Social Transformer (ST)

ST in our model integrates the motion tensors from CMT across all agents. By combining the individual representations from different agents, ST creates a comprehensive representation of the collective behavior, considering the influence and interactions among the agents. This enables the model to better understand and predict the complex dynamics in multi-agent scenarios.

$$SM_i = \mathbf{ST}(mH_i^M, i \in [1, N]).$$

In short, $\mathbf{ST}$ receives the motion cross-modal tensor of each agent $mH_i^M$ and provides a socially aware encoded tensor $SM_i$ with shape $(T_{pred}, e^T, D)$. We denote $SM_i = concat(SM_i^T, SM_i^Q)$, where $SM_i^T$ and $SM_i^Q$ are the mappings of $mH_i^T$ and $mQ_i$, respectively.

Finally, $SM_1^Q$ is processed through an MLP projection layer, outputting the predicted trajectory $Y$.

### 3.6 Masking

To ensure the generality and adaptability of our network, we employ a training approach that involves masking different types and quantities of visual cues. Each sample in the training dataset is augmented with a variable combination of cues, including trajectories, 2D or 3D pose keypoints, and bounding boxes. We introduce 1) modality-masking, where a specific visual cue is randomly masked across all time-steps, 2) meta-masking, which randomly masks a portion of the features for a given modality at some time-steps. This masking technique, applied to each sample, enables our network to learn and adapt to various cue configurations.

By systematically varying the presence or absence of specific cues in the input, we improve the model's ability to leverage different cues for accurate trajectory prediction. Our model is trained with Mean Square Error (MSE) loss function between $Y$ and ground-truth $\hat{Y}$.

## 4 Experiments

In this section, we present the datasets, metrics, baselines, and an extensive analysis of the results in both quantitative and qualitative aspects followed by the discussion. The implementation details will be found in Appendix A.6.

## 4.1 DATASETS

We evaluate on three publicly available datasets providing visual cues: the JTA (Fabbri et al., 2018) and the JRDB (Martin-Martin et al., 2021) in the main text, and the Pedestrians and Cyclists in Road Traffic (Kress et al., 2022) in Appendix A.1. Furthermore, we report on the ETH-UCY dataset (Pellegrini et al., 2009; Lerner et al., 2007), that does not contain visual cues in Appendix A.2.

**JTA dataset:** a large-scale synthetic dataset containing 256 training sequences, 128 validation sequences, and 128 test sequences, with a total of approximately 10 million 3D keypoints annotations. The abundance of data and multi-agent scenarios in this dataset enables a thorough exploration of our models' potential performance. We predict the location for the future 12 time-steps given the previous 9 time-steps at 2.5 frames per second (fps). Due to the availability of different modalities, this dataset is considered as our primary dataset, unless otherwise specified.

**JRDB dataset:** a real-world dataset that provides a diverse set of pedestrian trajectories and 2D bounding boxes, allowing for a comprehensive evaluation of our models in both indoor and outdoor scenarios. We predict the locations for the next 12 time-steps given the past 9 time-steps at 2.5 fps.

**Pedestrians and Cyclists in Road Traffic dataset:** gathered from real-world urban traffic environments, it comprises more than $2,000$ pedestrian trajectories paired with their corresponding 3D body pose keypoints. It contains $50,000$ test samples. For evaluations on this dataset, the models observe one second and predict the next 2.52 seconds at 25 fps.

## 4.2 METRICS AND BASELINES

We evaluate the models in terms of Average Displacement Error (ADE), Final Displacement Error (FDE), and Average Specific Weighted Average Euclidean Error (ASWAEE) (Kress et al., 2022):

- ADE/FDE: the average / final displacement error between the predicted locations and the ground-truth locations of the agent;

- ASWAEE: the average displacement error per second for specific time-steps; Following Kress et al. (2022), we compute it for these five time-steps: [t=0.44s, t=0.96s, t=1.48s, t=2.00s, t=2.52s].

We select the best-performing trajectory prediction models (Alahi et al., 2016; Kothari et al., 2021; Giuliari et al., 2021; Gupta et al., 2018) from the TrajNet++ leaderboard (Kothari et al., 2021). In addition, we compare with recent state-of-the-art models: EqMotion (Xu et al., 2023), Autobots (Girgis et al., 2022), Trajectron++ (Salzmann et al., 2020), and a pose-based trajectory prediction model (Kress et al., 2022). Note that in this paper, we concentrate on deterministic prediction, and thus, all models generate a single trajectory per agent. We refer to the deterministic version of Social-GAN (Gupta et al., 2018) as Social-GAN-det.

## 4.3 QUANTITATIVE RESULTS

Table 1 compares the previous models with our generic model on the JTA and JRDB datasets. First, we observe that our model, even when provided with only past trajectory information at inference time, surpasses previous models in terms of ADE / FDE. Nonetheless, the integration of pose keypoints information into our model leads to a significant enhancement. This improvement could stem from the ability of pose-informed models to capture body rotation patterns before changes in walking direction occur. The results also show that 3D pose yields better improvements compared to 2D pose. It can be attributed to the fact that modeling social interactions requires more spatial information, and 3D pose provides the advantage of depth perception compared to 2D pose.

The absence of pose keypoints information in the JRDB dataset led us to rely on bounding boxes. The results show that incorporating bounding boxes is better than solely relying on trajectories for accurate predictions. We conducted a similar experiment on the JTA dataset and observed that the inclusion of 2D bounding boxes, in addition to trajectories, improved the performance. However, it is important to note that the performance was still lower compared to utilizing 3D pose.

Furthermore, we conducted an experiment taking as input trajectory, 3D pose and 3D bounding box. The findings show that the performance of this combination was similar to using only trajectory and

Table 1: **Quantitative results on the JTA and JRDB datasets.** For each dataset, our model is trained once on all modalities with masking and evaluated using various combinations of visual cues at inference time ('T', 'P', and 'BB' abbreviate Trajectory, Pose, and Bounding Box, respectively.). Not availability of an input modality on a dataset is indicated with **/**.

| Models | Input Modality at inference | JTA dataset | | JRDB dataset | |
|---|---|---|---|---|---|
| | | ADE | FDE | ADE | FDE |
| Social-GAN-det (Gupta et al., 2018) | T | 1.66 | 3.76 | 0.50 | 0.99 |
| Transformer (Giuliari et al., 2021) | T | 1.56 | 3.54 | 0.56 | 1.10 |
| Vanilla-LSTM (Alahi et al., 2016) | T | 1.44 | 3.25 | 0.42 | 0.83 |
| Occupancy-LSTM (Alahi et al., 2016) | T | 1.41 | 3.15 | 0.43 | 0.85 |
| Directional-LSTM (Kothari et al., 2021) | T | 1.37 | 3.06 | 0.45 | 0.87 |
| Dir-social-LSTM (Kothari et al., 2021) | T | 1.23 | 2.59 | 0.48 | 0.95 |
| Social-LSTM (Alahi et al., 2016) | T | 1.21 | 2.54 | 0.47 | 0.95 |
| Autobots (Girgis et al., 2022) | T | 1.20 | 2.70 | **0.39** | 0.80 |
| Trajectron++ (Salzmann et al., 2020) | T | 1.18 | 2.53 | 0.40 | 0.78 |
| EqMotion (Xu et al., 2023) | T | 1.13 | 2.39 | 0.42 | 0.78 |
| Social-Transmotion | T | **0.99** | **1.98** | 0.40 | **0.77** |
| Social-Transmotion | T + 3D P | 0.89 | 1.81 | / | / |
| Social-Transmotion | T + 2D P | 0.95 | 1.91 | / | / |
| Social-Transmotion | T + 2D BB | 0.96 | 1.91 | **0.37** | **0.73** |
| Social-Transmotion | T + 3D P + 3D BB | 0.89 | 1.81 | / | / |
| Social-Transmotion | T + 3D P + 2D P + 3D BB + 2D BB | **0.88** | **1.80** | / | / |

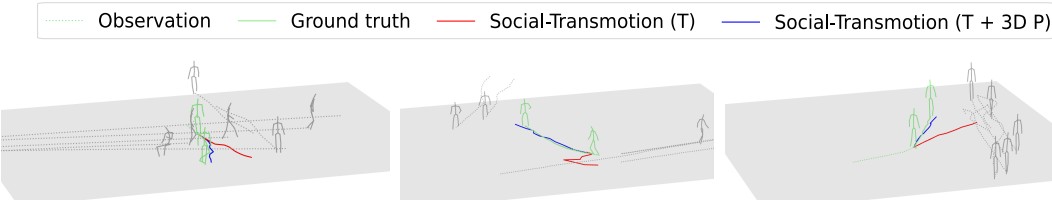

Figure 3: **Qualitative results showing the benefit of 3D pose input modality on trajectory predictions on the JTA dataset.** Red trajectories show predictions from Social-Transmotion with only trajectory (T) as the input modality, while blue trajectories depict predictions when both of the trajectory and 3D pose (T + 3D P) input modalities are used. The ground-truth are in green and the neighboring agents are in gray. The visualization includes the last observed pose keypoints to convey walking direction and body rotation.

3D pose as inputs. This suggests that, on average, incorporating 3D bounding box does not provide additional information beyond what is already captured from 3D pose.

Lastly, we assessed the model's performance using all accessible cues: trajectory, 3D and 2D poses, and 3D and 2D bounding boxes, which yielded the best outcomes.

## 4.4 QUALITATIVE RESULTS

Figure 3 provides a visual comparison between the Social-Transmotion model's outputs when utilizing solely trajectory as input, and when leveraging both trajectory and 3D pose information. The inclusion of pose keypoints enables the model to predict changes in the agent's direction and to avoid collisions with neighbors. For instance, in the left or the right figure, adding pose information allows the model to better understand body rotations and social interactions, resulting in predictions that are closer to the ground-truth and avoid collisions.

Predicting sudden turns presents a significant challenge for traditional trajectory prediction models. However, the incorporation of pose keypoints information can effectively address this issue. As demonstrated in the middle figure, the pose-informed model excels in scenarios involving sudden turns by leveraging pose data to anticipate forthcoming changes in walking state — an aspect the conventional model often fails to capture.

We detail some failure cases of the model in Appendix A.4.

Table 2: **Comparing the performance of the generic and specific models.** The generic model, trained once on all modalities with masking, is compared to six specific models trained only on modalities identical to those used at inference time in terms of ADE / FDE.

| Input Modality at inference | Generic Model | Specific Models |
|---|---|---|
| T | 0.99 / 1.98 | 1.00 / 2.04 |
| T + 3D P | 0.89 / 1.81 | 0.92 / 1.90 |
| T + 2D P | 0.95 / 1.91 | 0.97 / 1.99 |
| T + 2D BB | 0.96 / 1.91 | 0.99 / 1.96 |
| T + 3D P + 3D BB | 0.89 / 1.81 | 0.94 / 1.94 |
| T + 3D P + 3D BB + 2D P + 2D BB | 0.88 / 1.80 | 0.91 / 1.85 |

Table 3: **Evaluating the performance in imperfect input data settings.** It displays ADE / FDE and the corresponding performance degradation percentages compared to perfect input data situation in parenthesis. The generic model is trained on all visual cues with masking and the specific model is trained on the trajectory and 3D pose.

| Input Modality at inference | Generic Model
ADE / FDE (degradation% ↓) | Specific Model
ADE / FDE (degradation% ↓) |
|---|---|---|
| 100% T + 100% 3D P | 0.89 / 1.81 | 0.92 / 1.90 |
| 90% T + 90% 3D P | 0.89 / 1.81 (0.0% / 0.0%) | 1.27 / 2.48 (38.0% / 30.5%) |
| 50% T + 50% 3D P | 1.01 / 2.00 (13.5% / 10.5%) | 2.82 / 4.96 (206.5% / 161.1%) |
| 50% T + 10% 3D P | 1.10 / 2.16 (23.6% / 19.3%) | 3.06 / 5.48 (232.6% / 188.4%) |
| T + 3D P w/ Gaussian Noise (std=25) | 0.98 / 1.94 (10.1% / 7.2%) | 1.16 / 2.31 (26.1% / 21.6%) |
| T + 3D P w/ Gaussian Noise (std=50) | 1.05 / 2.05 (18.0% / 13.3%) | 1.35 / 2.68 (46.7% / 41.1%) |

## 4.5 DISCUSSIONS

**What if we use specific models?** Table 2 provides a comprehensive comparison between our proposed generic model and specific models. In this context, the generic model is trained once using all modalities with the masking strategy, whereas specific models are trained solely on modalities identical to the modalities used at inference time. The results indicate that the generic model consistently outperforms the specific models, underscoring the effectiveness of our masking strategy in improving the model's comprehension of visual cues.

**What if we have imperfect input?** In real-world scenarios, obtaining complete trajectories and body pose keypoints can often be hindered by obstructions or inaccuracies in pose estimation. To study this, we conducted an experiment evaluating the model with randomly masked trajectories and pose keypoints. We assessed the performance of the generic model (trained with all visual cues and masking) and a specific model (trained only on trajectories and 3D pose), as detailed in Table 3.

The results reveal that the generic model is significantly more robust against both low-quantities and low-quality input data. For instance, when faced with challenging scenarios of largely incomplete trajectory and pose input (50% T + 10% 3D P), the degradation in ADE / FDE for the generic model (23.6% / 19.3%) is markedly lower compared to the specific model (232.6% / 188.4%). Additionally, we observe that the generic model proves to be more adept at handling noisy pose keypoints than the specific model. By adding modality-masking and meta-masking techniques, our generic model reduces its dependence on any single modality and achieves greater robustness.

**What if we use different variations of 3D pose?** To explore how different variations of pose keypoints information contribute to accuracy improvements, we conducted a study. Initially, we analyzed whether performance enhancements were solely due to the primary agent's pose or if interactions with neighboring agents also played a significant role. Table 4 illustrates the substantial benefits of incorporating the primary agent's pose compared to relying only on trajectory data. Yet, performance further improved when the poses of all agents were included, underscoring the importance of considering social interactions with pose.

Next, we examined the impact of using only the last observed pose of all agents, along with their trajectories. The results indicate that relying solely on the last pose yields a performance comparable to using all time steps, emphasizing the critical role of the most recent pose in trajectory prediction.

Table 4: **Ablation studies on different variations of 3D pose.** The complete input situation is compared with where 1) only the primary agent's pose data is accessible, 2) pose information is restricted to the final time-step, 3) exclusively head pose data is available, and 4) pose is inaccessible.

| Models | Input Modality | ADE / FDE (degradation% ↓) |
|---|---|---|
| Social-Transmotion | T + 3D P | 0.89 / 1.81 |
| Social-Transmotion | T + 3D P (only primary's pose) | 0.92 / 1.91 (3.4% / 5.5%) |
| Social-Transmotion | T + 3D P (only last obs. pose) | 0.94 / 1.91 (5.6% / 5.5%) |
| Social-Transmotion | T + 3D P (only head pose) | 0.98 / 1.98 (10.1% / 9.4%) |
| Social-Transmotion | T | 0.99 / 1.98 (11.2% / 9.4%) |

Table 5: **Ablation studies on the model architecture.** 'CMT-ST' denotes the proposed model utilizing CMT followed by ST. 'MLP-ST' involves substituting CMT with MLP, 'ST-CMT' represents the reverse configuration, and 'CMT' signifies the usage of a single CMT only.

| Model Architecture | Input Modality | ADE / FDE (degradation% ↓) |
|---|---|---|
| CMT–ST | T + 3D P | 0.92 / 1.90 |
| MLP–ST | T + 3D P | 1.06 / 2.13 (15.2% / 12.1%) |
| ST–CMT | T + 3D P | 1.13 / 2.32 (22.8% / 22.1%) |
| CMT | T + 3D P | 1.23 / 2.80 (33.7% / 47.4%) |

Furthermore, our study considered the impact of using only head keypoints as visual cue, effectively excluding all other keypoints. According to the table, relying exclusively on head pose and trajectory yields performance similar to using trajectory alone, suggesting that the comprehensive inclusion of keypoints is needed for enhanced performance.

For further investigation, we provide spatial and temporal attention maps in Appendix A.3.

**What if we use other architectural designs instead of CMT–ST?** Our Social-Transmotion architecture, employing two Transformers—one for individual agents' features and another for agents interactions—was compared against three alternative architectures in Table 5.

In the MLP–ST design, we adopt a single-Transformer model, where a MLP extracts features of each agent, and the resultant tokens are aggregated with the transformer ST. However, the performance decline noted in the table underlines the CMT's superior feature extraction capability.

Further, we examined the effect of reversing the order of CMT and ST, focusing first on capturing the features of all agents at each time-step, and then using another Transformer to attend to all time-steps. This alteration led to increased errors, suggesting the importance of capturing an agent's feature over time before capturing all agents interactions. Our hypothesis is that the relationships between an agent's keypoints across different time-steps are more significant than the interactions among keypoints of multiple agents within a time-step and the ST–CMT approach presents a challenge to the network, as it must extract useful information from numerous irrelevant connections.

Lastly, a trial with only CMT, excluding ST, resulted in notable performance degradation, reaffirming the necessity of dual-Transformers for effective trajectory prediction.

## 5 CONCLUSIONS

In this work, we introduced Social-Transmotion, the first generic promptable Transformer-based trajectory predictor adept at managing diverse visual cues in varying quantities, thereby augmenting trajectory data for enhanced human trajectory prediction. Social-Transmotion, designed for adaptability, highlights that with an efficient masking strategy and a powerful network, integrating visual cues is never harmful and, in most cases, helpful (free win). By embracing multiple cues on human behavior, our approach pushed the limits of conventional trajectory prediction performance.

**Limitations:** While our generic model can work with any visual cue, we have examined a limited set of visual cues and noted instances in the appendix where they did not consistently enhance trajectory prediction performance. In the future, one can study the potential of alternative visual cues such as gaze direction, actions, and other attributes, taking into account their presence in datasets.

## 6 ACKNOWLEDGEMENT

The authors would like to thank Yuejiang Liu, Jifeng Wang, and Mohammadhossein Bahari for their valuable feedback. This work was supported by Sportradar[1] (Yang's Ph.D.), the Swiss National Science Foundation under the Grant 2OOO21-L92326, and Valeo AI[2].

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

## A APPENDIX

### A.1 PERFORMANCE ON THE PEDESTRIANS AND CYCLISTS IN ROAD TRAFFIC DATASET

Table 6 compares our model to Kress et al. (2022) that used 3D body pose keypoints to predict human trajectories on this dataset. The notations 'c' and 'd' represent two variations of their model using a continuous or discrete approach, respectively. The results indicate the effectiveness of our dual Transformer and its proficiency in utilizing pose information due to the masking strategy.

Table 6: **Quantitative results on the Pedestrians and Cyclists in Road Traffic dataset.** Models are compared in terms of ASWAEE (the lower, the better).

| Models | Input Modality | ASWAEE $\downarrow$ |
|---|---|---|
| $c_{traj}$ (Kress et al., 2022) | T | 0.57 |
| $d_{traj}$ (Kress et al., 2022) | T | 0.60 |
| Social-Transmotion | T | **0.44** |
| $c_{traj,pose}$ (Kress et al., 2022) | T + 3D P | 0.51 |
| $d_{traj,pose}$ (Kress et al., 2022) | T + 3D P | 0.56 |
| Social-Transmotion | T + 3D P | **0.40** |

### A.2 PERFORMANCE ON THE ETH-UCY DATASET

The ETH-UCY dataset (Pellegrini et al., 2009; Lerner et al., 2007) is a widely recognized dataset providing pedestrian trajectories in a birds-eye view, without other types of visual cues. It comprises five subsets: ETH, Hotel, Univ, Zara1 and Zara2. Following established conventions in previous research, our model predicts 12 future locations based on 8 past ones, observed at 2.5 fps.

Table 7 demonstrates the deterministic prediction performance of our model compared to previous works. Notably, our model shows commendable performance on the challenging ETH subset, and maintains competitive results on other subsets using solely trajectory data as input. This is attributed to the efficacy of our dual-Transformer architecture.

Table 7: **Quantitative results on the ETH-UCY dataset.** Models are compared in ADE / FDE of deterministic predictions. The reported numbers of previous works are from their original papers.

| Models | ETH | Hotel | Univ | Zara1 | Zara2 | Average |
|---|---|---|---|---|---|---|
| Social-GAN-det (Gupta et al., 2018) | 1.13/2.21 | 1.01/2.18 | 0.60/1.28 | 0.42/0.91 | 0.52/1.11 | 0.74/1.54 |
| Social-LSTM (Alahi et al., 2016) | 1.09/2.35 | 0.79/1.76 | 0.67/1.40 | 0.47/1.00 | 0.56/1.17 | 0.72/1.54 |
| Transformer (Giuliari et al., 2021) | 1.03/2.10 | 0.36/0.71 | 0.53/1.32 | 0.44/1.00 | 0.34/0.76 | 0.54/1.17 |
| MemoNet (Xu et al., 2022) | 1.00/2.08 | 0.35/0.67 | 0.55/1.19 | 0.46/1.00 | 0.32/0.82 | 0.55/1.15 |
| Trajectron++ (Salzmann et al., 2020) | 1.02/2.00 | 0.33/0.62 | 0.53/1.19 | 0.44/0.99 | 0.32/0.73 | 0.53/1.11 |
| Autobots (Girgis et al., 2022) | 1.02/1.89 | 0.32/0.60 | 0.54/1.16 | 0.41/0.89 | 0.32/0.71 | 0.52/1.05 |
| EqMotion (Xu et al., 2023) | 0.96/1.92 | **0.30/0.58** | **0.50/1.10** | **0.39/0.86** | **0.30/0.68** | **0.49/1.03** |
| Social-Transmotion | **0.93/1.81** | 0.32/0.60 | 0.54/1.16 | 0.42/0.90 | 0.32/0.70 | 0.51/**1.03** |

### A.3 ATTENTION MAPS

To explore the impact of different keypoints and time-steps on trajectory prediction, the attention maps are displayed in Figure 4. The first map shows temporal attention, while the second illustrates spatial attention. The attention weights assigned to earlier frames are comparatively lower, suggesting that later frames hold more critical information for trajectory prediction. In simple scenarios, the last observed frame may suffice, as evidenced by our previous ablation study. However, in complex scenarios, a larger number of observation frames might be necessary. We also found that specific keypoints, such as the ankles, wrists, and knees, play a crucial role in determining direction and movement. There is generally symmetry across different body points, with a slight preference towards the right, which may reflect a bias in the data. These insights pave the way for further research, particularly in identifying a sparse set of essential keypoints that could provide benefits in specific applications.

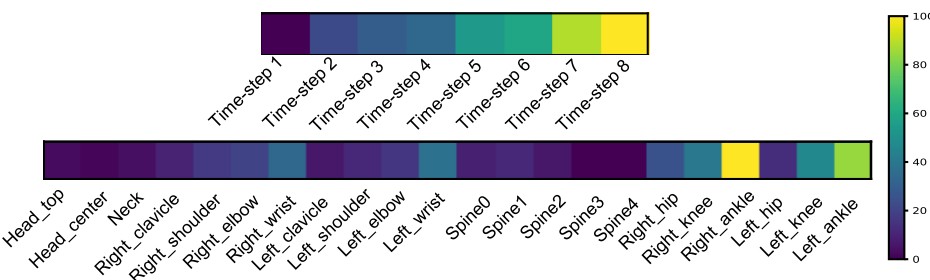

Figure 4: **Temporal (top) and spatial (bottom) attention maps:** These maps underscore the significance of particular time-steps and body keypoints in trajectory prediction.

In Figures 5 and 6, we present two examples involving turns. In the simpler scenario (Figure 5), a single time-step that captures body rotation suffices. Conversely, for the more complex scenario (Figure 6), multiple frames prove to be informative in facilitating accurate trajectory prediction.

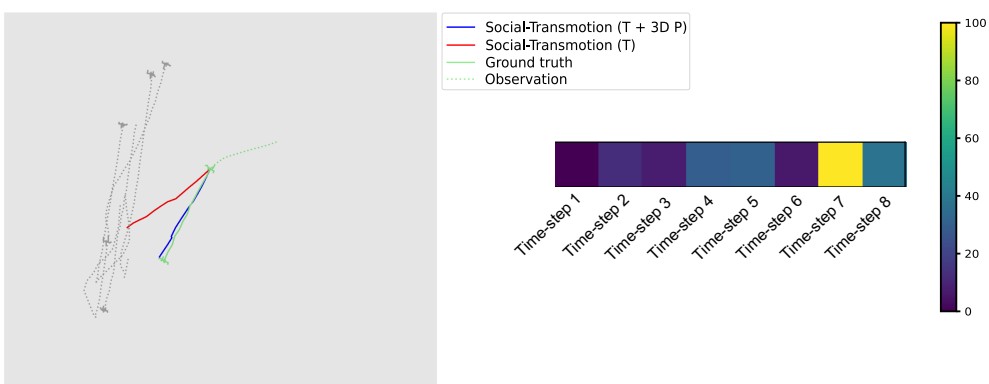

Figure 5: **Qualitative example of a simple turning scenario:** This temporal attention map illustrates how a single key frame can be pivotal.

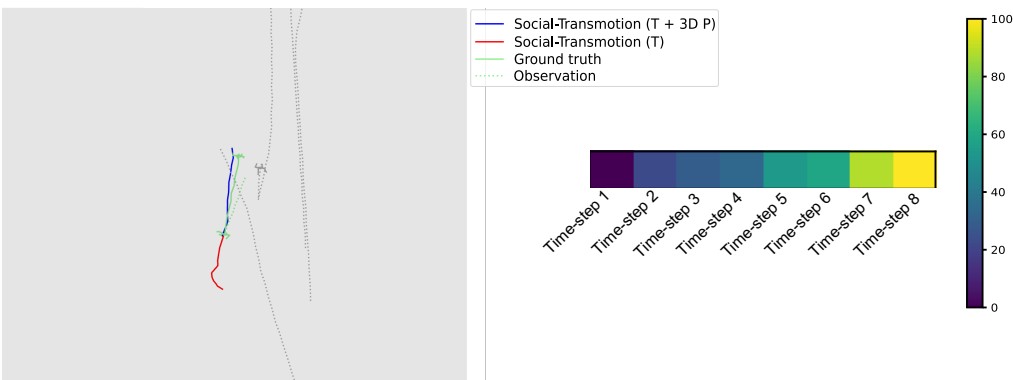

Figure 6: **Qualitative example of a challenging turning scenario:** This temporal attention map shows the importance of multiple frames in complex scenarios.

## A.4 QUALITATIVE RESULTS: FAILURE CASES

We have included illustrative examples showcasing instances where the model's performance falls short. These examples provide valuable insights and identify areas for potential improvement. For instance, as portrayed in Figure 7 and Figure 8, relying solely on pose keypoints may not always

yield optimal outcomes. The integration of supplementary visual cues like gaze or the original scene image could potentially offer advantageous improvements.

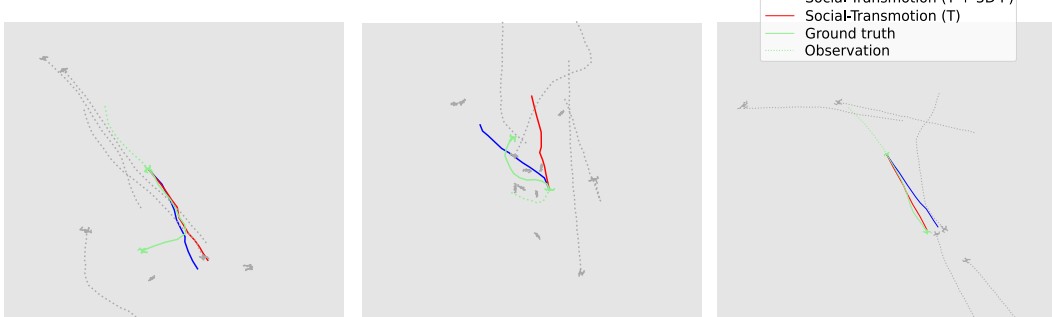

Figure 7: **Some failure cases.** For the primary agent, the ground-truth trajectory, trajectory-informed prediction and pose-informed predictions are in green, red, and blue, respectively. All other agents are depicted in gray. The pose of the last observed frame is also visualized to indicate walking direction and body rotation. The efficacy of pose information varies; the left figure demonstrates an inevitable scenario where the individual alters their path in the middle of the future horizon. The middle figure shows a crowded scene, and the right one shows a situation where pose information offers limited value.

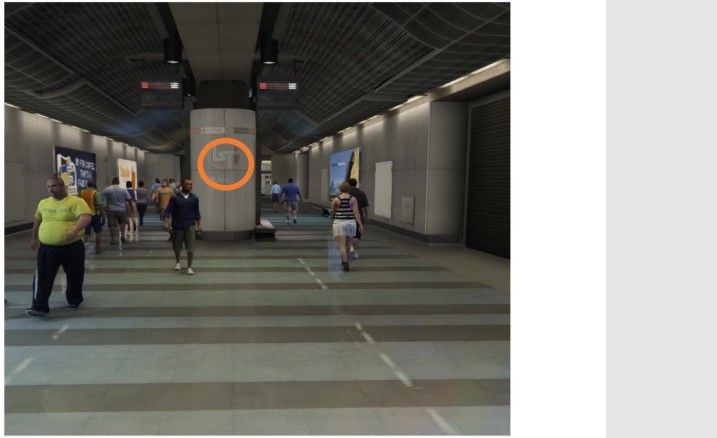 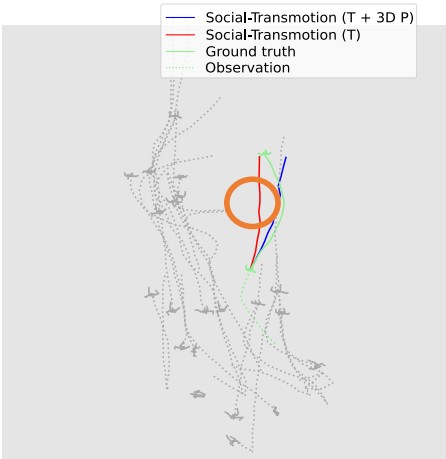

Figure 8: **A failure case due to missing context.** This instance underscores the dependence of our model on abstract information, which resulted in the inability to account for scene barriers. Incorporating scene features could mitigate such issues.

### A.5 IMPERFECT POSE

Following our experiments on the model performance in noisy conditions, we extend the study to include scenarios where ground-truth pose information is unavailable. We conducted an experiment on the JRDB dataset using an off-the-shelf pose estimator (Grishchenko et al., 2022) and then using the estimated 3D poses as input cues to evaluate our model without re-training. As illustrated in Table 8, our model demonstrates a notable ability to utilize even imperfect pose estimations, leading to improved performance compared to using only trajectories as input.

Although the performance gain with pseudo-ground-truth pose on the JRDB dataset is slightly lower compared to the gains seen with actual ground-truth pose on the JTA dataset, this study underscores our model's adaptability and robustness in real-world applications where accurate ground-truth data may be difficult to obtain. To potentially close this performance gap, one approach could be training the model directly with the estimated or inaccurate poses.

Table 8: **The performance of Social-Transmotion using trajectory and estimated 3D pose from an off-the-shelf estimator.** The reported numbers are ADE / FDE.

| Input Modality | ADE/FDE |
|---|---|
| T | 0.40 / 0.77 |
| T + estimated 3D P | 0.36 / 0.72 |

Moreover, we conducted experiments to assess our model's performance under occlusions by masking keypoints in specific temporal or spatial patterns. These experiments included: 1) Random Leg and Arm Occlusion: 50% of arm and leg keypoints randomly occluded; 2) Structured Right Leg Occlusion: all right leg keypoints consistently occluded; 3) Complete Frame Missing: all keypoints in selected frames missing at a set probability. The results presented in Table 9, affirm the model's robustness against both temporal and spatial occlusions, with diminishing performance only at very high occlusion rates. Additionally, Figure 9 qualitatively illustrates that the model performs comparably with full or partially occluded keypoints.

Table 9: **Model's robustness evaluation under various occlusion scenarios**, measured by ADE/FDE. Scenarios include random, structured, and complete frame occlusions with different probabilities, demonstrating the model's efficacy across different levels of occlusion.

| Input Modality at inference | ADE / FDE |
|---|---|
| T + Clean 3D Pose | 0.89 / 1.81 |
| T + Random Leg and Arm Occlusion | 0.90 / 1.83 |
| T + Structured Right Leg Occlusion | 0.90 / 1.82 |
| T + Complete Frame Missing (50%) | 0.93 / 1.89 |
| T + Complete Frame Missing (90%) | 0.98 / 1.96 |
| T + Complete Frame Missing (100%) | 0.99 / 1.98 |

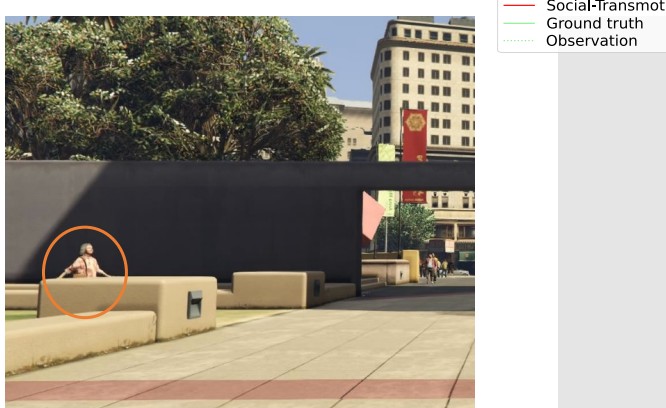 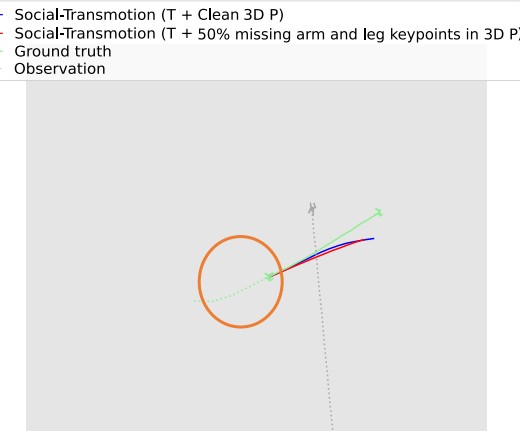

Figure 9: **Qualitative demonstration of model performance with complete vs occluded keypoint inputs.** This illustrates that the model effectively handles occlusions, showing similar performance even with 50% of the visual cues missing.

## A.6 IMPLEMENTATION DETAILS

The architecture of CMT includes six layers and four heads, whereas ST is constructed with three layers and four heads; both utilize a model dimension of 128. We employed the Adam optimizer (Kingma & Ba, 2014) with an initial learning rate of $1 \times 10^{-4}$ , which was reduced by a factor of 0.1 after 80% of the 50 total epochs were completed. We had 30% modality-masking and

10% meta-masking. All computations were performed on a NVIDIA V100 GPU equipped with 32GB of memory.

For the JRDB dataset, after extracting data, we used the TrajNet++ (Kothari et al., 2021) code base to generate four types of trajectories with acceptance rates of '1.0, 1.0, 1.0, 1.0'. We used 'gates-ai-lab-2019-02-08_0' for validation, the indoor video 'packard-poster-session-2019-03-20_1' and the outdoor video 'bytes-cafe-2019-02-07_0', 'gates-basement-elevators-2019-01-17_1', 'hewlett-packard-intersection-2019-01-24_0', 'huang-lane-2019-02-12_0', 'jordan-hall-2019-04-22_0', 'packard-poster-session-2019-03-20_2', 'stlc-111-2019-04-19_0', 'svl-meeting-gates-2-2019-04-08_0', 'svl-meeting-gates-2-2019-04-08_1', and 'tressider-2019-03-16_1' for training.

For the JTA dataset, after extracting data, we performed pose normalization and then processing with the TrajNet++ (Kothari et al., 2021) preprocessor to generate four types of trajectories with acceptance rates of '0.01, 0.1, 1.0, 1.0'. The full list of data splits can be found in our code repository.

