# OpenReview forum: "Social-Transmotion: Promptable Human Trajectory Prediction"
_ICLR.cc/2024/Conference — ICLR 2024 poster_

### Official Review · Reviewer_KGKG · 2023-10-29

**Soundness:** 3 good
**Presentation:** 3 good
**Contribution:** 3 good
**Rating:** 6
**Confidence:** 4

**Summary:**

- The paper introduces Social-Transmotion, a model for human trajectory prediction leveraging transformer architectures to process diverse visual cues.
- The model innovatively utilizes the concept of a "prompt" from Natural Language Processing, which could be x-y coordinates, bounding boxes, or body poses, to augment trajectory data.
- Social-Transmotion is adaptable to various visual cues and employs a masking technique to ensure effectiveness even when certain cues are unavailable.
- The paper investigates the importance of different keypoints and frames of poses for trajectory prediction, and the merits of using 2D versus 3D poses.
- The model's effectiveness is validated on multiple datasets, including JTA, JRDB, Pedestrians and Cyclists in Road Traffic, and ETH-UCY.

**Strengths:**

- The idea of prompting human trajectory prediction seems novel to me. Incorporating (optional) bounding box sequences and/or 2D/3D sequences makes sense, which would likely lower prediction errors and also be useful in real-world applications. The proposed framework could also be potentially scalable to other prompts (e.g. video features, scenes, etc.)
- This paper is in general well-written, with adequate experiments to support the claim.

**Weaknesses:**

- I like the motivation of 'What if we have imperfect input?'. Nonetheless, to better support the claim, authors could consider more realistic input artifacts (e.g. use real detectors, masks for detection failures, etc.) in addition to Gaussian noises. The notation of '-188.4%' is somewhat confusing; using '+' should be fine.
- Qualitative comparison with regard to existing baselines would be beneficial for better understanding the performance improvement.

**Questions:**

This work uses multiple prompts but only decodes trajectories. Can the authors also discuss how to explore predicting finer body motions (pose) and the relationship with regard to previous works in multi-person motion prediction?

---

> ### Author Response · Authors · 2023-11-22
>
> Thank you for your valuable feedback. Below are the responses to your comments.
>
> > Utilizing pose estimates from real detectors.
>
> In response to your interesting suggestion, we conducted an experiment involving an off-the-shelf pose estimator [1], on the JRDB dataset (ground-truth 3D pose is not available). The obtained estimated 3D poses were then used as input cues for our model. As demonstrated in the table below, our model exhibits a notable capacity to leverage even imperfect pose estimations, resulting in a performance improvement.
> Note that while the gain achieved with pseudo-ground-truth pose (here on the JRDB dataset) is slightly lower compared to utilizing ground-truth pose (as showcased in the main paper with a gain of about 10.1\% and 8.58\% on the JTA dataset), the model's adaptability to real-world pose estimators underscores its robustness and practical utility in scenarios where accurate ground-truth information may be challenging to obtain.
>
> |                    | T         | T + estimated 3d P      |
> |--------------------|-----------|-------------------------|
> | Social-Transmotion | 0.40/0.77 | 0.36/0.72 (8.70%/6.80%) |
>
> As you mentioned, in table 2 of the paper, we evaluated the performance using different percentages of available cues and introduced Gaussian noise, providing a comprehensive analysis of the model's robustness under various conditions. We add this discussion to the paper too.
>
> > Qualitative comparison with baselines.
>
> Following your suggestion, in the appendix, we have provided visual comparisons between our model and the best baseline, EqMotion, on the JTA dataset. Both models successfully capture interactions with neighboring agents. However, our model exhibits a higher level of progressiveness, leveraging interactions across a greater number of agents and more modalities (pose and trajectory). This distinct feature results in predictions that are closer to the ground truth.
>
>
> > Investigating multi-person full-body motion prediction.
>
> While our primary focus in the paper centered around the trajectory prediction task, it opens future explorations into extending our model to pose prediction. In response to your comment, we conducted a preliminary experiment, modifying our model to predict poses as a new output head.
> In this initial exploration, given the constraints of time, we focused on the single-person pose prediction task using the Human3.6M dataset. Following the settings of EqMotion (from CVPR 2023), we assessed the prediction results in terms of Mean Per Joint Position Error (MPJPE) across different prediction horizons, as presented in the below table.
>
> | Model                                  | 80ms | 160ms | 320ms | 400ms |
> |----------------------------------------|------|-------|-------|-------|
> | EqMotion                               | 9.1  | 20.1  | 43.7  | 55.0  |
> | Social-Transmotion + pose decoder head | 9.2  | 21.4  | 48.5  | 61.2  |
>
> The findings indicate that although our model performs well in pose prediction, it currently does not outperform the state-of-the-art. This underscores the need for further refinement and optimization in future investigations.
>
>
> > Table notations change.
>
> Thanks. We changed the notations in the tables, and the PDF will be updated accordingly.
>
> [1] Grishchenko et al, Blazepose ghum holistic: Real-time 3d human landmarks and pose estimation, CVPRW 2022

---

### Official Review · Reviewer_qMC2 · 2023-10-31

**Soundness:** 2 fair
**Presentation:** 3 good
**Contribution:** 2 fair
**Rating:** 5
**Confidence:** 5

**Summary:**

The paper presents a novel approach for pedestrian trajectory prediction, focusing on incorporating various visual cues such as 3D poses and bounding boxes, in addition to the traditional trajectory information. The model, named Social-Transmotion, utilizes a dual-transformer architecture to effectively integrate these visual cues and enhance the prediction accuracy, especially in challenging scenarios involving interactions among pedestrians.

The results demonstrate that the inclusion of 3D poses significantly improves the model's performance, outperforming other state-of-the-art models and various ablated versions of itself. The model also shows robustness against incomplete or noisy input data, highlighting its practical applicability in real-world scenarios.

**Strengths:**

- Incorporation of Visual Cues: The model effectively utilizes additional visual cues like 3D poses and bounding boxes, which is a significant advancement over traditional trajectory-only models.

- Robustness: The model demonstrates robust performance even when faced with incomplete or noisy input data, showcasing its reliability for real-world applications.

- Performance: Social-Transmotion outperforms various state-of-the-art models and its own ablated versions, indicating its effectiveness in pedestrian trajectory prediction.

**Weaknesses:**

- Lack of Commonly Used Benchmarks: Some commonly used datasets are not used such as nuScenes, Agroverse 1/2, Waymo Open Motion Dataset, etc. These datasets are often used to evaluate the performance of trajectory prediction methods.

- Complexity: The inclusion of various visual cues and a dual-transformer architecture might make the model computationally intensive, potentially limiting its applicability in resource-constrained environments.

- Dependence on Accurate Pose Estimation: The model's performance is significantly enhanced by the inclusion of 3D poses, which necessitates accurate pose estimation. Inaccuracies in pose estimation could potentially degrade the model's performance.

- Limited Exploration of Failure Cases: While the paper mentions the provision of failure cases in the appendix, a more thorough exploration and discussion of these cases within the main text could provide valuable insights for further improvements.

- Missing Relevant Recent Baselines: [1-4] are some recent methods that are relevant to this work.
[1] Uncovering the Missing Pattern: Unified Framework Towards Trajectory Imputation and Prediction, CVPR 2023
[2] Query-Centric Trajectory Prediction, CVPR 2023
[3] Unsupervised Sampling Promoting for Stochastic Human Trajectory Prediction. CVPR 2023
[4] AdamsFormer for Spatial Action Localization in the Future, CVPR 2023

**Questions:**

1. How does the computational complexity of Social-Transmotion compare to other state-of-the-art models, and what are the implications for its real-world applicability?

2. Could you elaborate on the model's performance in scenarios with inaccurate or noisy pose estimations, and what strategies could be employed to mitigate potential performance degradation?

3. Are there specific types of interactions or scenarios where Social-Transmotion particularly excels or struggles, and what insights can be drawn from these cases?

4. How does the model handle occlusions, and what is the impact on its performance when key visual cues are partially or fully obscured?
Could you provide more details on the failure cases mentioned, and what lessons were learned from these cases to further improve the model?

---

> ### Author Response · Authors · 2023-11-22
>
> Thank you for your insightful review. The following are our responses to the points you have raised.
>
> > The performance on other datasets: Argoverse, Waymo.
>
> In addition to the JTA dataset, our paper incorporated various common datasets in human trajectory prediction, including multi-person multi-modal robotic scenarios (JRDB), multi-person single-modal traffic scenarios (ETH-UCY), and single-person multi-modal traffic scenarios (Pedestrians and Cyclists in Road Traffic dataset). To enhance our analysis of multi-person multi-modal traffic scenarios and following your suggestion, we added the Waymo Perception dataset. This dataset was chosen for its extensive multi-modal annotations, including 2D/3D bounding boxes and 2D/3D pose keypoints of humans, as well as frame conversions and 2D-3D label associations.
> Due to the substantial size of the dataset and the time constraints, we conducted training and evaluation on selected subsets (25,000 random samples from the training set and 5,000 random samples from the validation set). Our model leverages the various visual labels and their associations provided by this dataset whenever they are available, and masks them out when not.
>
> For comparison, we selected two recent top-performing models based on the results presented in Table 1 of our paper: EqMotion (from CVPR 2023) and the transformer-based Autobots (from ICLR 2022). The detailed results can be found here:
>
> |                    | ADE/FDE   |
> |--------------------|-----------|
> | Autobots           | 0.40/0.85 |
> | EqMotion           | 0.44/0.88 |
> | Social-Transmotion | 0.39/0.82 |
>
> It shows that our model exhibits better performance compared to previous works. We acknowledge that with further hyperparameter tuning, optimization, and training on the entire dataset, our model's performance could be further enhanced. We have added this experiment to the paper.
>
> > Computational costs comparison.
>
> Thank you for the suggestion. We have analyzed the computational costs of our model with various visual cues in Appendix A4. Upon your recommendation, we now include a comparison with the Eqmotion and Autobots baselines in the table below, detailing inference times based on over 5000 samples using a Tesla V100 GPU. Notably, our dual-transformer model achieves faster inference speeds than these baselines, even when processing 3D Pose data, which is advantageous for resource-limited environments. This efficiency gain is partly due to our use of CMT for encoding modalities and information compression, allowing the subsequent transformer, ST, to operate on a compacted learned representation.
>
> |                     | EqMotion | Autobots | Social-Transmotion (T) | Social-Transmotion (T+3d P) |
> |---------------------|----------|----------|------------------------|-----------------------------|
> | Inference time (ms) | 8.28     | 7.50     | 5.74                   | 6.73                        |
>
> It is important to acknowledge that our approach relies on additional estimation methods within its pipeline. Hence, when considering the complete processing time, especially for potential real-world deployments, these factors should be taken into account. However, these estimation methods have significantly improved in speed recently, enabling real-time usage without substantial concerns.
>
>
> > How's the model's performance with inaccurate or noisy pose estimations?
>
> Regarding the performance with less accurate or noisy pose estimations, we refer to the first part of the Discussions in Section 4.4, titled "What if we have imperfect input?". There, in table 2 of the paper, we evaluated the performance using different percentages of available cues and noisy pose, providing an analysis of the model's robustness under various conditions.
>
> In response to you and another reviewer, we conducted another complementary experiment involving an off-the-shelf pose estimator [1], on the JRDB dataset (ground-truth 3D pose is not available). The obtained estimated 3D poses were then used as input cues for evaluating our model without re-training. As demonstrated in the table below, our model exhibits a notable capacity to leverage even imperfect pose estimations, resulting in a performance improvement.
>
> |                    | T         | T + estimated 3d P        |
> |--------------------|-----------|---------------------------|
> | Social-Transmotion | 0.40/0.77 | 0.36/0.72 (8.70\%/6.80\%) |
>
> Note that while the gain achieved with pseudo-ground-truth pose (here on the JRDB dataset) is slightly lower compared to utilizing ground-truth pose (as showcased in the main paper with a gain of about 10.1\% and 8.58\% on the JTA dataset), our generic model's adaptability to real-world pose estimators underscores its robustness and practical utility in scenarios where accurate ground-truth information may be challenging to obtain.
> One way to lower this gap is to train the model with the estimated/inaccurate poses.

---

> > ### Author Response · Authors · 2023-11-23
> >
> > > The model's performance with partially or fully-occluded pose
> >
> > In Table 3 of the paper, we have highlighted the significance of individual poses, such as the head pose, in comparison to the full-body pose. Additionally, in the previous paragraph of the response, we've considered the impact of employing a real-world pose estimator that may encounter occlusions.
> > Following your suggestion, we conducted experiments to assess our model's performance under occlusions by masking keypoints in specific temporal or spatial areas. These experiments included these scenarios:
> >
> > 1) Random leg and arm occluded 3d P: leg and arm joints are randomly occluded with the same probability of $50\%$;
> > 2) Structurally occluded 3d P: the right leg joints for all frames are missing;
> > 3) Whole frame missing 3d P: the pose in some frames are completely missing with a dedicated probability;
> >
> > Our findings in the table below demonstrate that our model maintains robustness against both temporal and spatial occlusions, with diminishing performance only at very high occlusion rates.
> >
> > | Occlusion in temporal and spatial space              | ADE/FDE   |
> > |------------------------------------------------------|-----------|
> > | T + Complete 3d P                                    | 0.89/1.81 |
> > | T + Random leg and arm occluded 3d P                 | 0.90/1.83 |
> > | T + Structurally occluded 3d P                       | 0.90/1.82 |
> > | T + Whole frame missing 3D P (with 50\% probability) | 0.93/1.89 |
> > | T + Whole frame missing 3D P (with 80\% probability) | 0.98/1.96 |
> > | T + Fully occluded pose (no 3d P)                    | 0.99/1.98 |
> >
> > Furthermore, in the appendix we showed one qualitative example on how inputting all keypoints yields similar results compared to giving only occluded keypoints.
> >
> > > More discussions on successful and failure cases.
> >
> > As shown in the main paper (successful cases), Social-Transmotion, engineered for adaptability, underlines that integrating visual cues are never harmful and, in most cases helpful (free win) but still there are some limitations to be addressed in the future.
> > For instance the failure cases in the appendix have shown that limited set of visual cues do not consistently enhance trajectory prediction performance. In the future, one can study the potential of alternative visual cues such as gaze direction, actions, and other attributes, taking into account their presence in datasets. These cues could potentially provide a more comprehensive understanding of human mobility patterns, leading to more precise predictions of human trajectories.
> > Another intriguing avenue for research that we mentioned in the paper involves benefiting directly from images. This could facilitate the transformation of images into optimized prompts, enabling the direct utilization of visual information.
> >
> > The experiments also revealed that imperfect observation such as excessive noise in pose data (e.g., Gaussian noise with a standard deviation of 50) could misguide the model, pointing to a possible limitation. To mitigate this, one option is to do data augmentation techniques, which might improve robustness at the expense of accuracy on clean data. The choice would be application-dependent. Alternatively, selecting and developing a more reliable pose estimator could address this issue. This expanded discussion has been included in the manuscript.
> >
> > > Adding four relevant works.
> >
> > Thanks. We have added a section in our manuscript discussing vehicle trajectory prediction and have cited the works you mentioned.

---

### Official Review · Reviewer_mJke · 2023-11-02

**Soundness:** 3 good
**Presentation:** 2 fair
**Contribution:** 3 good
**Rating:** 5
**Confidence:** 5

**Summary:**

The paper presents a transformer based approach for motion prediction. It focus on using visual cues alongside agent location to predict the future. The future can be position, pose or bounding box. The future is made into a prompt along side the visual cues.  A cross modality transformer is used to combine the different modalities the another transformer is used for motion prediction.  The reported results are good in comparison with previous work.

**Strengths:**

- The model can handle different "types" of motion predictions [pose, position, bounding box]
- The selective masking techniques which can be seen as a form of data balancing is being employed for better results
- The latent input is a nice approach for these problems, similar encoding can be found in [1] where an encoder was used to generate a codebook.
- The discussion section is rich. For example the analysis of imperfect data with degradation percentage is valuable for the domain. The masking behavior is similar to the work of [2].



[1] MotionGPT: Human Motion as a Foreign Language
[2] Deep Tracking: Seeing Beyond Seeing Using Recurrent Neural Networks

**Weaknesses:**

- In the introduction, it was mentioned that "traditional predictors have limited performance, as they typically rely on a single data point per person (i.e., their x-y coordinates on the ground) as input." This can not be a general statement as works such as [1] and following ones do tackle the point using such cues.  Also, it seems this work is not mentioned in the related work section.
- In the related work section there need to be a balance between the 3 modes supported in the work, where there is a literatures for each mode with different directions.
- Figure 2 doesn't show the path for the visual cues mentioned in 3.2. Or there is a confusion between the word "visual" and "spatial" cue?
Did the authors mean spatial cues such as pose, bounding box or visual cues such as the partial image of the scene beside the spatial cues?





[1] SoPhie: An Attentive GAN for Predicting Paths Compliant to Social and Physical Constraints

**Questions:**

- I'm strongly wondering about the ADE/FDE results. The proposed model output is deterministic where most of the method reported in the table are probabilistic except Social-LSTM. I'm only aware of [[1]-appendix c] where there is an approach to compare deterministic and probabilistic models. What is the authors comment on this?

- Another suggestive study, the impact of training data amount on the model performance. It seems from section 3.2 that data imperfection might impact the performance. What about the impact of data quantity? like using 10%, 20% ... etc on the results?

- The confusion between naming "visual" and "spatial" cues is impacting the readability/expectations of the article and need to be addressed.

[1]Social-Implicit: Rethinking Trajectory Prediction Evaluation and The Effectiveness of Implicit Maximum Likelihood Estimation

---

> ### Author Response · Authors · 2023-11-22
>
> Thank you for your helpful input. Please find our answers to the raised points below:
>
> > Ensuring a balanced related work section for all supported prediction modes.
>
> We would like to clarify that our work does not encompass multiple prediction modes; rather, it is exclusively focused on trajectory prediction. Consequently, in our related works section, we have a subsection dedicated to the human trajectory prediction task and another subsection addressing the ways in which the utilization of visual cues can enhance it. These input modalities are not predicted but utilized to augment trajectory prediction accuracy. We have endeavored to cite all relevant studies on the use of each input modalities and are grateful for any additional references we may have overlooked.
>
> > Deterministic vs. probabilistic ADE/FDE?
>
> Indeed, as you pointed out, some of the models in the main table had a probabilistic approach and some deterministic.
> Notably, some of the probabilistic models have implemented a deterministic scenario and have reported ADE/FDE metrics for both deterministic and probabilistic results (e.g., EqMotion, Trajectory Transformer). To facilitate consistency in reporting, we have adjusted the prediction mode to 1 for probabilistic models (as they provided in their opensource public codes), asking them to predict the most reliable trajectory as the deterministic scenario and report deterministic ADE/FDE metrics.
>
> > Performance changes with different data quantity in training.
>
> We appreciate your recommendation to investigate the impact of varying amount of data in training on the performance. Following your suggestion, we conducted experiments with models trained on incrementally increasing data quantities. The outcomes of these experiments, as detailed below, indicate a marked performance enhancement up to the 40\% data volume threshold.
> Post this threshold, we observe steady improvements up until the 80\% data mark. Post-80\%, there is a noticeable plateau in performance enhancement.
>
> | Training data quantity | 10\%      | 20\%      | 40\%      | 60\%      | 80\%      | 100\%     |
> |------------------------|-----------|-----------|-----------|-----------|-----------|-----------|
> | ADE/FDE                | 1.20/2.41 | 1.14/2.32 | 1.01/2.06 | 0.95/1.95 | 0.90/1.85 | 0.88/1.80 |
>
> We added this experiment to the paper.
>
>
> > Visual cues or spatial cues?
>
> In our manuscript, we used the term "visual cues" to refer to information extracted visually, such as the coordinates of 2D pose keypoints appear within the image frames or . Explicitly in this paper we refer to visual cues, whether they are poses, bounding boxes, or a combination thereof. We recognize the potential for ambiguity and have therefore added the definition in the manuscript. The exploration of additional input cues, including but not limited to scene context, is put for future research.
>
>
> > Not all traditional predictors have limited performance, because of reliance on a single data point per person as input.
>
> Thanks for pointing it out. We wanted to express that single-modality input is a limitation for predictors that relies solely on x-y coordinates, not for all of them. We updated the manuscript and added the missed citation.

---

### Official Review · Reviewer_BUFd · 2023-11-02

**Soundness:** 2 fair
**Presentation:** 3 good
**Contribution:** 2 fair
**Rating:** 3
**Confidence:** 5

**Summary:**

The paper presents another take on transformers being applied to the field of trajectory prediction. The authors claim the novelty of the approach being the inclusion of other visual cues into the transformer framework like 2D, 3D bounding boxes and keypoints. The paper reports results on various academic datasets and presents ablation on various input modalities.

**Strengths:**

1) The paper is well written and easy to understand
2) The paper presents good ablative analysis based of the input modalities that are used for the task of trajectory prediction.
3) The paper evaluates results on various publically available datasets.

**Weaknesses:**

1) The paper lacks novelty as the use of transformers using multi-modal inputs for the task of trajecory prediction is already been studied extensively including works like Wayformer for example.
2) The paper does not use larger industrial-academix datasets like Argoverse or Waymo open motion dataset to compile results against other transformer based benchmarks popular in the field today.

Given the above two major weaknesses, even with the nice experimental section and exhasutive results it is difficult to see how this work adds value to the field.

**Questions:**

It would be great to compare the architecture against more relevant transformer based baselines on industrial level datasets.

---

> ### Author Response · Authors · 2023-11-22
>
> Thank you for your comments. Our responses to each point are outlined below.
>
> > Using transformers for trajectory prediction lacks novelty, echoing established studies like Wayformer.
>
> As we mentioned in the paper, transformers have been widely utilized recently in trajectory prediction.
> However, it is essential to emphasize a distinctive contribution in our study. Our primary innovation lies in the introduction of a generic model with our modality-masking technique. This technique enhances the generality and robustness of the model, enabling flexible utilization of visual cues  (we train **only one** model with all the available cues and at test time generate predictions only based on a subset of these cues), as demonstrated in Table 1 of our manuscript.
>
> Regarding the architecture, our proposed dual-transformer is modified from the language model BERT, which is different from the encoder-decoder structure used in Wayformer.
> The main similarity of our model and Wayformer for example, is that we both use latent queries to encode features from different modalities.
> However, in the case of Wayformer and most trajectory prediction architectures, the modalities represent only the locations of agents and map information. Our model differs in its focus on broader visual cues beyond that, leading to distinct learning processes for agent interactions.
>
> In Wayformer, the model learns the agent-agent interaction based on the location. In contrast, our model, designed for the task of human trajectory prediction, necessitates learning not only location-wise interactions but also pose interactions across agents. This distinction becomes particularly crucial in complex, crowded scenes where each agent may exhibit multiple modalities. Consequently, to address the challenge of modeling these intricate interactions, we employ distinct transformers for cross-modal feature extraction and social interactions, which enables the model to learn how neighbours' poses, bounding boxes and trajectories affect the agent's future trajectory.
>
> > Additional datasets like Argoverse and Waymo for benchmarking against other transformer-based models.
>
> In addition to the JTA dataset, our paper incorporated various common datasets in human trajectory prediction, including multi-person multi-modal robotic scenarios (JRDB), multi-person single-modal traffic scenarios (ETH-UCY), and single-person multi-modal traffic scenarios (Pedestrians and Cyclists in Road Traffic dataset). To enhance our analysis of multi-person multi-modal traffic scenarios and following your suggestion, we added the Waymo Perception dataset. This dataset was chosen for its extensive multi-modal annotations, including 2D/3D bounding boxes and 2D/3D pose keypoints of humans, as well as frame conversions and 2D-3D label associations.
> Due to the substantial size of the dataset and the time constraints, we conducted training and evaluation on selected subsets (25,000 random samples from the training set and 5,000 random samples from the validation set). Our model leverages the various visual labels and their associations provided by this dataset whenever they are available, and masks them out when not.
>
> For comparison, we selected two recent top-performing models based on the results presented in Table 1 of our paper: EqMotion (from CVPR 2023) and the transformer-based Autobots (from ICLR 2022). The detailed results can be found here:
>
> |                    | ADE/FDE   |
> |--------------------|-----------|
> | Autobots           | 0.40/0.85 |
> | EqMotion           | 0.44/0.88 |
> | Social-Transmotion | 0.39/0.82 |
>
> It shows that our model exhibits better performance compared to previous works. We acknowledge that with further hyperparameter tuning, optimization, and training on the entire dataset, our model's performance could be further enhanced. We have added this experiment to the paper.

---

### Meta-Review · Area_Chair_eNyB · 2023-12-10

**Metareview:**

Synopsis: This paper tackles the problem of human trajectory prediction using a new transformer-based learned model, which is prompted with a variety if input cues. The paper includes promising results on several state of the art datasets.

Strengths:
+ The ability to accept a variable set of input types is very useful, and the model masking idea is novel and interesting.
+ The paper includes results from three existing datasets.
+ The addition of the analysis of the impact of occlusions on the results is valuable

Weaknesses:
- A more thorough analysis on existing AV datasets would help place the proposed work's accuracy in context of the state of the art.
- The analysis of impact of real detectors on the results could be more thoroughly investigated

**Justification For Why Not Higher Score:**

The empirical evaluation seems to be "above the bar", but could be made more thorough. The reviews include specific suggestions for improvement, which are non-trivial, but which are necessary for evaluating an algorithm in this established and crowded field.

**Justification For Why Not Lower Score:**

The ideas are still interesting. It would be great to have this work presented.

---

### Decision · Program_Chairs · 2024-01-16

Accept (poster)